# Deep Reinforcement Learning Based Freshness-Aware Path Planning for UAV-Assisted Edge Computing Networks with Device Mobility

**Yingsheng Peng** [1] , **Yong Liu** [1,*], **Dong Li** [2] **and Han Zhang** [1]

1   School of Electronics and Information Engineering, South China Normal University, Foshan 528225, China
2   Faculty of Information Technology, Macau University of Science and Technology, Macau 999078, China
*   Correspondence: yliu@m.scnu.edu.cn; Tel.: +86-1840-712-5369

**Abstract:** As unmanned aerial vehicles (UAVs) can provide flexible and efficient services concerning the sparse network distribution, we study a UAV-assisted mobile edge computing (MEC) network. To satisfy the freshness requirement of IoT applications, the age of information (AoI) is incorporated as an important performance metric. Then, the path planning problem is formulated to simultaneously minimize the AoIs of mobile devices and the energy consumption of the UAV, where the movement randomness of IoT devices are taken into account. Concerning the dimension explosion, the deep reinforcement learning (DRL) framework is exploited, and a double deep Q-learning network (DDQN) algorithm is proposed to realize the intelligent and freshness-aware path planning of the UAV. Extensive simulation results validate the effectiveness of the proposed freshness-aware path planning scheme and unveil the effects of the moving speed of devices and the UAV on the achieved AoI.

**Keywords:** unmanned aerial vehicle; mobile edge computing; path planning; deep reinforcement learning; age of information

## 1. Introduction

With the development of the Internet of Things (IoT) and fifth-generation cellular communication, task computing has undergone a paradigm shift from centralized cloud to mobile edge computing [1]. As a promising technology, mobile edge computing (MEC) can alleviate the tension between computing-intensive applications and mobile devices with limited resources [1,2]. The MEC offers the computing capability at the network edge in close proximity to end devices, i.e., radio access network, such that the large distance transmission is no longer necessary, and the information is generated locally and consumed locally. Thus, the quality of service is improved due to a small energy consumption and execution latency [3,4]. The traditional MEC framework with static deployment may not be efficient in serving the devices with sparse distribution and mobility [5]. Instead, the unmanned aerial vehicle (UAV) assisted MEC framework has been recently exploited to enlarge the service coverage and improve performance [6]. While, how to optimize the path trajectory of the UAV becomes a critical issue to achieve these benefits.

The MEC architecture equipped with UAVs can improve the connectivity of ground wireless devices and expand the coverage due to its mobility [7]. Thus, it has attracted much research attention [7–11]. The improvement of computing performance introduced by the UAV-assisted MEC framework is validated in [7]. In [10], the trajectory of the UAV and bit allocation scheme are jointly optimized subject to the transmission delay and the UAV's energy constraints. In [11], beside the trajectory, the user association is optimized to satisfy the quality of service (QoS) requirement of applications by minimizing the total energy consumption of the UAV. All these works focus on optimizing the path planning and the resource allocation for the UAV-assisted MEC networks to achieve a good performance in terms of network throughput, delay, and service reliability. Nowadays, as

more and more real-time IoT applications are emerging, how to capture the freshness of the information becomes a significant requirement, which has not been fully investigated in the above works.

To realize timely information transmission and computing, the age of information (AoI) that characterizes how old the information is from the perspective of the receiver has been recently proposed as an important performance metric and has attracted much research attention [12–14]. The achieved AoI of IoT devices is studied in [15–18] by optimizing the UAV trajectory. In [17], the achieved AoI is minimized for UAV-assisted wireless powered IoT network. For a UAV-assisted IoT network, the age-optimal data collection problem is studied in [18] to minimize the weighted sum of AoI, which is formulated as a finite time Markov decision process (MDP). In [19], the UAV is employed with computational capability and acts as an edge server to reduce the power consumption and AoI of IoT devices. While, the UAV trajectory planning is not taken into account, and the IoT devices are assumed to be static. In practice, the location of the device may change dynamically over time. In general, the dynamic moving of devices inevitably causes the huge or countless states of devices' positions, and this results in the path planning problem becoming very complicated. To address it, the deep reinforcement learning (DRL) framework is suitable to deal with the massive dimensions of states and actions caused by mobile devices [11]. Motivated by the above observations, this paper focuses on optimizing the service trajectory of UAVs for a mobile edge computing network to optimize the information freshness of IoT devices.

Different from most of the existing works, the mobility of IoT devices is taken into account in the path planning problem of an energy-constrained UAV in this paper. Meanwhile, when the UAV is moving to an access point, multiple devices in the communication area can be associated and upload their computing task to the UAV. A path planning problem is formulated to minimize the sum of AoI of all mobile devices and the energy consumption of the UAV. Then, a DRL algorithm is proposed to design the flying trajectory of the UAV intelligently. The main contributions of this paper are summarized as following:

- We study a UAV-assisted MEC network with device mobility, in which an energy-constrained UAV as a relay station collects and forwards the computing task of mobile devices within its serviceable area to a faraway base station to be remotely executed.
- The path planning problem is formulated to simultaneously minimize the energy consumption of the UAV and the averaged AoI by optimizing the service path of the UAV. Considering the dimension explosion issue caused by the enormous state space, a freshness-aware path planning scheme based on double deep Q-learning network (DDQN) algorithm is proposed to optimize the trajectory of the UAV intelligently.
- Extensive experiments are conducted to validate that the proposed freshness-aware path planning scheme performs better than the conventional schemes. Meanwhile, the effects of the moving speed of the UAV and mobile devices on the achieved AoI are unveiled. We further present the example of devices' AoI evolution and the UAV's service trajectory.

The remainder of this paper is organized as follows. The related works are presented in Section 2. The system model and the problem formulation are described in Section 3. The deep reinforcement learning-based freshness-aware path planning scheme is presented in Section 4. Experiments and Results are provided in Section 5, followed by discussion and conclusion in Section 7. All symbols used in this paper are listed in Table 1.

**Table 1.** Notations and Definitions.

| Notation | Definition |
|---|---|
| $N$ | Number of IoT devices |
| $T_s$ | Communication time of the UAV |
| $l_c$ | Number of bits in a task |
| $p$ | Probability of task generation |
| $m(t)$ | Indicate whether the UAV is flying or hovering |
| $(x_n(t), y_n(t))$ | Position of device $n$ in the time slot $t$ |
| $(x_0(t), y_0(t))$ | Position of the UAV in the time slot $t$ |
| $d_n(t)$ | The distance between device n and UAV in the time slot $t$ |
| $H$ | The height of the UAV from the ground |
| $\alpha$ | The maximum angle from the UAV and the edge of its coverage area |
| $u_n(t)$ | Indicate whether device n is in the association area in the time slot $t$ |
| $\eta_n(t)$ | Indicate the active status of device n in the time slot $t$ |
| $\lambda(t)$ | The amount of unloaded task at the time slot $t$ |
| $P_f$ | Flight power of UAV |
| $L$ | The flying distance of the UAV |
| $V$ | The flying speed of a UAV |
| $P_h$ | Hovering power of UAV |
| $B$ | System bandwidth |
| $N_0$ | Noise power spectral density |
| $\rho_0$ | Path loss per meter |
| $(x_B, y_B)$ | The position of the BS |
| $t_u$ | Relay forwarding time |
| $E(t)$ | The energy status of the UAV in The time slot $t$ |
| $Z_n(t)$ | The AoI of device n in the time slot $t$ |
| $E_0$ | Initial energy of UAV |
| $\Delta_n(t)$ | The time interval of two successive computing tasks generated |

## 2. Related Works

The number of smart IoT devices grows rapidly, and it is estimated that there will be 25 billion Internet of things devices by 2025 [20], which leads to the emergence of diverse applications, i.e., real-time video analysis [21], virtual reality [22], and smart city [23]. These applications are computationally intensive and delay-sensitive, depending on our ability to process data and extract useful information quickly [5]. To support these applications, mobile edge computing is regarded as a promising network architecture, which enables cloud computing power and information technology (IT) service environment at the network edge. By pushing data-intensive tasks to the edge and processing data locally in neighboring MEC servers, this architecture has the potential to significantly reduce latency, avoid congestion, and prolong the lifetime of mobile devices [20,24]. While, in the case of massive devices or sparse distribution of network facilities, the existing MEC technology is not applicable.

With the advantages of wide coverage, fast deployment, and strong scalability, the UAV-assisted framework has been proposed to serve the ground mobile devices. Various function modules, such as sensors, small base station, embedded computing server, can be integrated into the UAV to achieve large-scale sensing, communication, and computing [25–27]. A four-layer hierarchical network composed of sensor nodes, cluster heads, UAV, and BS was studied in [27] and a simulated annealing algorithm was applied to design the trajectory of the UAV. UAV can be used as a wireless relay station or air base station, using its maneuverability to fly closer to each mobile device to improve the connectivity of ground wireless devices and expand coverage [28]. In [10], the UAV trajectory and bit allocation scheme were jointly optimized by minimizing the total consumed energy subject to the quality of service (QoS) requirements of IoT applications. A UAV-assisted MEC network was explored in [11], where the UAV executed the computational tasks offloaded from devices. The total consumed energy was minimized by optimizing the device association and UAV trajectory. In [28], both the service delay and the weighted sum of consumed

energy of the UAV were minimized by optimizing the UAV location, communication and computing resource allocation, and task offloading decision. All these works focused on the path planning and resource management in UAV-assisted communication networks via the traditional optimization methods, and the great potential of UAV-assisted communication was validated. While, the communication environment becomes complex as the dynamic movement characteristic, which causes a great challenge in addressing the path planning and resource management problem in the UAV-assisted communication problem.

Recently, as an exciting field of artificial intelligence (AI), the deep reinforcement learning (DRL) framework has been exploited to the sequenced decision-making problem in diverse applications. Due to the explosive growth of states/actions, the traditional methods in the form of table storage become infeasible. Instead, DRL can effectively approach the Q-value of reinforcement learning via a deep neural network (DNN) [29]. Thus, DRL has been widely used in online resource allocation and scheduling design in wireless networks [4,30–32]. A computational offloading algorithm based on DDQN was proposed for a MEC network in [30], in which mobile devices learned the task offloading and energy allocation scheme based on the queue status and channel quality of the task and energy. [31] studied the dynamic caching, computational offloading, and resource allocation in cache-assisted multi-user MEC systems with random task arrivals. A dynamic scheduling strategy based on a deep deterministic policy gradient method based on DRL was proposed. The spectrum sharing problem in the vehicle network was formulated as a multi-agent reinforcement learning problem in [32], and a multi-agent deep Q-network (DQN) algorithm was proposed. All these works have validated that the DRL framework is effective for the optimization problem of communication networks.

Nowadays, real-time IoT applications are emerging and timely information updates/communication play a significant role in improving the QoS. To achieve it, a new performance metric, AoI, has recently been proposed to evaluate the interval between the current time and the generation time of the most recently received packet at the destination [33–35]. Afterward, the AoI was extended to the UAV-assisted framework, namely AoI-oriented UAV communication networks [36–38]. Based on the DRL framework, an online flying trajectory of the UAV was designed in [36] to achieve the minimum and weighted sum of AoIs. In [37], the consumed energy and flying trajectory of the UAV was optimized to achieve the minimum peak AoI, where the UAV acted as a relay server. The AoI-oriented UAV path planning and data acquisition problems are jointly optimized in [38]. Some researchers have used MEC technology to optimize the AoI of IoT devices and have made some progress recently [12,39,40]. In [39], they studied the AoI of MEC's computationally intensive messages and verified that the average AoI of local computing was the smallest compared with local computing and remote computing. In [12], the channel allocation and computational offloading decision were jointly optimized to reduce the system cost while guaranteeing the freshness requirement of sensory data. In [40], the authors designed a system that supports MEC for the Internet of medical things, which minimized the system cost of medical emergencies, AoI, and energy consumption. In the UAV-assisted mobile edge computing network, considering AoI is also a current issue worthy of research. In [19], the authors proposed a UAV-assisted MEC system, and the freshness of data packets is studied separately in the case of single and multiple IoT device transmission. While, this work does not take into account the UAV's trajectory planning and it assumes that the IoT devices are static.

Based on the above observations, this paper focuses on designing the service trajectory for a UAV-assisted MEC network to minimize the information freshness of IoT devices. Meanwhile, the dynamic movements of IoT devices are taken into account.

## 3. System Model and Problem Formulation

### 3.1. System Model

We study a UAV-assisted MEC network as illustrated in Figure 1, in which the communication time is slotted with a duration of $T_s$ second. $N$ mobile devices are randomly

deployed in a communication area for diverse IoT applications, i.e., environment sensing, monitoring, and so on. Such that some specific computing tasks are generated at these devices to be further gathered and processed by a faraway base station (BS). We assume the tasks of mobile devices generate randomly. When the task queue of the device is empty, it may generate new tasks in the one time slot with probability $p$, where the task size is $l_c$ bits. Meanwhile, each device is employed with a single task queue to store the generated tasks. As the limit of local computation capability resources at the devices, the generated computing tasks have to be offloaded to the BS for remote execution, where the BS is employed with an edge server.

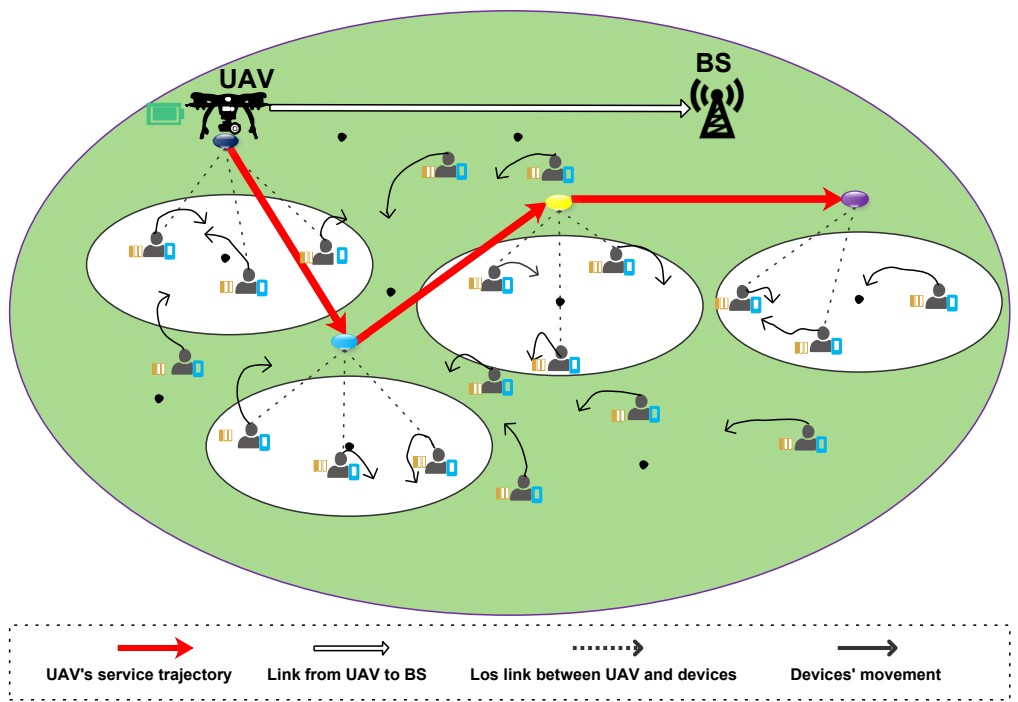

**Figure 1.** The system model of UAV-assisted edge computing networks consists of a UAV and multiple mobile devices.

The direction transmission from devices to the BS is blocked as the large communication distance and channel attenuation. To address it, a UAV is deployed as a mobile relay station to enlarge the communication coverage and assist the information delivery. Accordingly, the transmission from devices to the BS is divided into two phases. In the first phase, the UAV dynamically flies to one of the access points and serves the associated devices. Then, in the second phase, the UAV forwards the collected information to the BS via the wireless backhaul. In each time slot, the UAV is either flying to or hovering at an access point to serve devices. Thus, we use a binary variable $m(t) \in \{0,1\}$ to indicate whether the UAV is flying or hovering in the time slot $t$, and $m(t) = 1$ means the UAV is in flight. Due to the large distance between the access points, the UAV may spend multiple slots to fly from one access point to another. Followed by the flying period, the UAV is then hovering to collect the tasks from devices within one time slot, where the time is allocated to support the transmission of multiple devices. An example time structure of transmission scheme in a UAV-assisted MEC network is presented in Figure 2.

For the communication links, the ground-to-air line-of-sight (LoS) link model is assumed, which is widely used in UAV-assisted communication networking. Accordingly, the channel conditions mainly depend on the communication distances between the UAV and devices or BS, respectively, which means the accuracy of channel conditions relies on localization. In this paper, the UAV is enabled to acquire the channel condition of all communication links. It is sure that the imperfect channel state information (CSI) is

practical, and the resource management under imperfect CSI has been researched in the literature [41–43]. In [41], the performance of aerial computing supported by UAVs under non-ideal channel estimation is analyzed. In [42], the authors study an optimization framework for the energy-efficient transmission of NOMA-cooperative vehicle-to-everything (V2X) networks enhanced for ambient backscatter (AMBC) communication under imperfect CSI conditions. Even though the method and proposed scheme under perfect CSI can still provide some criterion guidance for the design under imperfect scenarios. Instead, we leave this part for our future works. All the notations and symbols used in this paper are listed in the Table 1.

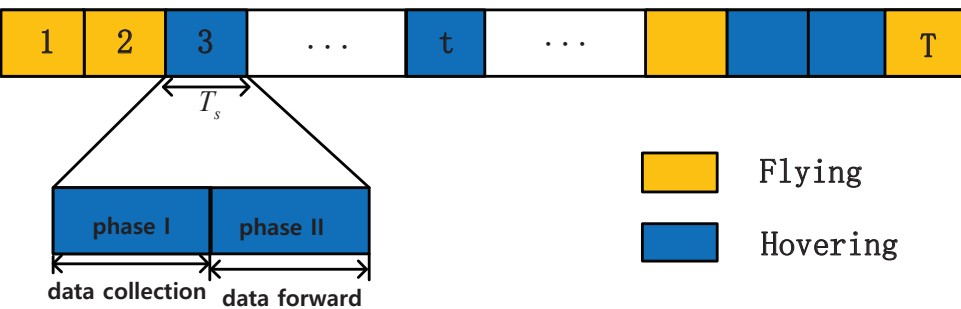

**Figure 2.** Time structure of transmission scheme.

### 3.2. Movement Model

Unlike the existing works with static deployment, this paper considers the devices can randomly move from one point to another during a time slot, namely mobile devices. The Gauss–Markov random movement model is adopted. We define $(x_n(t), y_n(t))$ as the position of device $n$ at the beginning of time slot $t$. Then, based on [44], the position of device $n$ in the time slot $t+1$ is updated as

$$x_n(t+1) = x_n(t) + v_n(t)\cos(\theta_n(t))T_s, \tag{1}$$

$$y_n(t+1) = y_n(t) + v_n(t)\sin(\theta_n(t))T_s, \tag{2}$$

where $v_n(t)$ and $\theta_n(t)$ are the moving speed and direction of device $n$ in the time slot $t$ respectively, and given by

$$v_n(t) = c_1 v_n(t-1) + c_2 v + \sqrt{1 - c_1^2}\,\Phi_n, \tag{3}$$

$$\theta_n(t) = d_1 \theta_n(t-1) + d_2 \theta_n + \sqrt{1 - d_1^2}\,\Psi_n. \tag{4}$$

In the above expressions, $v$ and $\theta_n$ represent the averaged moving speed and direction of device $n$ respectively, the parameters $c_1, c_2, d_1, d_2$ are used to evaluate the coherence of the devices' movement during successive time slots. Meanwhile, $\Phi_n$ and $\Psi_n$ are two random variables that follow independent Gaussian distributions, i.e., $\Phi_n \sim (\bar{\bar{\xi}}_{v_n}, \zeta_{v_n}^2)$, $\Psi_n \sim (\bar{\bar{\xi}}_{\theta_n}, \zeta_{\theta_n}^2)$. These two distributions reflect the randomness of the movement of different mobile devices.

### 3.3. Communication Model

To enlarge the coverage and assist in information delivery, the UAV dynamically flies in the communication area. We define $(x_0(t), y_0(t))$ as the position of the UAV in the time slot $t$. Accordingly, the communication distance between device $n$ and the UAV is

$$d_n(t) = \sqrt{[x_n(t) - x_0(t)]^2 + [y_n(t) - y_0(t)]^2}, \tag{5}$$

which determines the scheduling scheme and energy consumption of device $n$. A device can upload its task to the UAV only when its distance is no larger than the threshold of maximum allowable communication distance $d_0$. $d_0 = H tan\alpha$ relies on both the height of the UAV $H$, and the maximum angle between the UAV and the edge of its coverage area, which is dependent on the factor that the UAV is equipped with a directional antenna with adjustable beam width, and the azimuth and elevation half-power beam-width of the antenna are equal [45]. Thus, we can use a binary variable $u_n(t) \in \{0,1\}$ to denote whether the mobile device $n$ is in the association area of the UAV in the time slot $t$. If $d_n(t) \leq d_0$, it yields $u_n(t) = 1$, and the UAV can serve device $n$. In contrast, $u_n(t) = 0$, and device $n$ can not transmit information to the UAV as the large transmission distance, i.e.,

$$u_n(t) = \begin{cases} 1, & \text{if } d_n(t) \leq d_0, \\ 0, & \text{otherwise.} \end{cases} \tag{6}$$

In fact, a device is active to upload its computing task to the UAV in time slot $t$, only when it is in the association area of the UAV and its task queue is non-empty. Therefore, we define $\eta_n(t) \in \{0,1\}$ as the active status of device $n$ in the time slot $t$, then we have

$$\eta_n(t) = \begin{cases} 1, & \text{if } u_n(t) = 1 \text{ and non-empty queue,} \\ 0, & \text{otherwise.} \end{cases} \tag{7}$$

Accordingly, the number of active devices in the time slot $t$ is $\eta(t) = \sum_{n=1}^{N} \eta_n(t)$, and the set of active devices is defined as $\mathcal{N}_\eta(t) = \{n | \eta_n(t) = 1\}$.

In each time slot, an active device can only upload a computing task, which is forwarded by the UAV to the BS for edge execution. Therefore, the total amount of data bits that are sent to the UAV in the time slot $t$ is given by

$$\lambda(t) = \sum_{n=1}^{N} \eta_n(t) l_c, \tag{8}$$

which is also the amount of data bits forwarded by the UAV and affects the energy consumption of the UAV for the relaying operation.

### 3.4. Energy Consumption

In fact, the energy consumption of devices is much smaller than that of the UAV, which significantly affects the achieved performance of the whole system. Thus, we mainly investigate the energy consumption of the UAV, which is used for flying, hovering, and information relaying operations.

### 3.4.1. Flying Energy Consumption

During the flying period, the amount of consumed energy depends on the flying distance, speed, and power, which is given in [46] as

$$e_f(t) = m(t) P_f \frac{L}{V}, \tag{9}$$

where $L = \sqrt{[x_0(t) - x_0(t-1)]^2 + [y_0(t) - y_0(t-1)]^2}$ represents the flying distance of the UAV, $V$ and $P_f$ are the constant flying speed and power of the UAV's horizontal movement, respectively.

### 3.4.2. Hovering Energy Consumption

When the UAV hovers in an access point in the time slot $t$, it collects and forwards the information of active devices within the serviceable area. Therefore, both the hovering

and forwarding operation consume energy, in which the hovering energy consumption is written as [18]

$$e_q(t) = (1 - m(t))P_h T_s, \tag{10}$$

where $P_h$ is hovering power.

### 3.4.3. Relaying Energy Consumption

In the time slot $t$, if the UAV is hovering in an access point, e.g., $m(t) = 0$, the amount of information bits to be forwarded by the UAV is $\lambda(t)$. To achieve it, the transmit power of the UAV is

$$P_u(t) = \frac{\left(2^{\frac{\lambda(t)}{Bt_u}} - 1\right)BN_0}{l(t)}, \tag{11}$$

where $t_u$ and $\frac{\lambda(t)}{Bt_u}$ represent the forwarding transmission time and the normalized data rate, respectively, $B$ and $N_0$ denote the system bandwidth and noise power spectral density respectively. $l(t) = \frac{\rho_0}{\sqrt{H^2 + [x_0(t) - x_B]^2 + [y_0(t) - y_B]^2}}$ is the channel power gain between the UAV and BS, where $(x_B, y_B)$ is the position of the BS, and $\rho_0$ is the reference path loss of 1 meter. Then, the actual energy consumed for relaying operation is

$$e_u(t) = (1 - m(t))P_u(t)t_u. \tag{12}$$

Above all, the total energy consumption of the UAV in the time slot $t$ is

$$e_{total}(t) = e_f(t) + e_q(t) + e_u(t). \tag{13}$$

On this basis, we use $E(t)$ to denote the energy status of the UAV in the time slot $t$ and set the initial energy as $E(0) = E_0$. Accordingly, the energy status of the UAV is not only dependent on the energy status in the last time slot but also relies on the energy consumed in the current time slot. Thus, the evolution of the energy status of the UAV is given by

$$E(t + 1) = E(t) - e_{total}(t). \tag{14}$$

### 3.5. Age of Information

For real-time IoT applications, timely information updates/transmission plays a critical role in improving service quality. To realize timely data delivery, the performance metric called the age of information (AoI) has been recently proposed to evaluate the interval between the current time and the generation of the most recently received information packet of a device. In this paper, we incorporate the AoI as an important performance metric to achieve a freshness-aware UAV-assist MEC network.

The AoI characterizes how old the information is from the perspective of the receiver. Let's define $Z_n(t)$ as the AoI of device $n$ in the time slot $t$. According to the communication model, if the UAV is flying, devices can not upload their tasks to the UAV as $\eta_n(t) = 0$, then we have $Z_n(t + 1) = Z_n(t) + T_s$ as the information at the BS is one slot older. In contrast, when the UAV is hovering in an access point and device $n$ is active, i.e., $m(t) = 0, \eta_n(t) = 1$, the BS successfully receives a task from device $n$. Then, the AoI is degraded to $Z_n(t) + T_s - \Delta_n(t)$, where $\Delta_n(t)$ is the time interval of two successive computing tasks generated by device $n$. Thus, the evolution of the AoI of device $n$ is

$$Z_n(t + 1) = Z_n(t) + T_s - \eta_n(t)\Delta_n(t)(1 - m(t)). \tag{15}$$

Accordingly, the long-term average AoI of device $n$ is given by

$$\bar{Z}_n = \frac{1}{T}\sum_{t=1}^{T} Z_n(t). \tag{16}$$

*3.6. Problem Formulation*

In a UAV-assisted MEC network, we focus on the performance metrics of the energy consumption of the UAV and the achieved average AoIs of devices. Thus, our goal is to minimize the energy consumption of the UAV simultaneously and the averaged AoI by optimizing the service path of the UAV, and it can be formulated as the following optimization problem

$$\min_{x_0(t),y_0(t)} \quad \frac{1}{T} \sum_{t=0}^{T-1} \sum_{n=1}^{N} \omega_1 Z_n(t) + \frac{1}{T} \sum_{t=0}^{T-1} \omega_2 e_{total}(t) \tag{17}$$
$$s.t. \quad (1) - (16), \quad E(t) \geq 0,$$

where the first and second terms of the objective function are related to the averaged AoIs of devices and the averaged energy consumption of the UAV, and $\omega_1, \omega_2$ is the corresponding weighted coefficient. Noting that (17) is not only a non-convex optimization problem due to the non-convexity of the Equations (6) and (12), but also a long-term multi-slots optimization scheduling problem. Both factors increase the difficulty of solving the problem.

## 4. Deep Reinforcement Learning Dased Freshness-Aware Path Planning

The formulated problem (17) is to design the service trajectory of the UAV, which is like the sequenced decision-making issue. As the DRL framework is suitable to address such kind of decision-making issue. Thus, in this section, we explore the DRL framework to realize the intelligent and freshness-aware path planning of the UAV.

*4.1. Reinforcement Learning Reformulation*

Traditional tabular reinforcement learning, in which the tables are used to represent the states and the Q value of each state-action pair. In fact, the optimization problems formulated from the practical scenarios are complicated due to a huge number of states, such that the tabular reinforcement learning method is difficult to solve it. During the past years, reinforcement learning has made a significant breakthrough by exploiting a DNN to optimize decision-making, namely DRL. With the benefit of DNN, we can directly generate and approximate Q values instead of recording them in a table, which leads to a large decrease in computation complexity. Neural networks are usually employed to adapt to complex unknown functions in learning tasks [47], and they can handle complex input characteristics. In [48], the DRL framework is explored to train the CNN network, such that the online strategies are proposed to play computer games. Similarly, the original path planning optimization problem can be reformulated based on the DRL.

In a reinforcement learning framework, an agent continuously interacts with a complex and unknown environment, adjusts its policy, and learns to become an adaptive decision maker. In this paper, the UAV is an agent to explore the unknown external environment [49,50]. In the following parts, we firstly present the state space, action space, and reward function of the path planning optimization problem.

4.1.1. State Space

In time slot $t$, the UAV is either flying or hovering in an access point, namely $m(t) = 1$ or $m(t) = 0$. When the UAV is hovering and serving some devices in a time slot, it is necessary to decide the next hovering access point based on the environment state and its strategy. The environment states include the status of mobile devices, communication links, and the UAV. On the one hand, the environment states includes the current positions of devices $(x_n(t), y_n(t))$ and the coordinates of last time slot of UAV $(x_0(t-1), y_0(t-1))$, the channel fading of the communication link from the UAV to the BS $l(t)$, and the energy level of the UAV $E(t)$. On the other hand, to guarantee fresh information delivery, the AoI of mobile devices $Z_n(t)$ is also taken into account as an important state. Accordingly, the state is described as

$$S_t = \left\{ \ (x_n(t), y_n(t)), (x_0(t-1), y_0(t-1)), l(t), Z_n(t), E(t) \ \right\}, \tag{18}$$

and $\mathcal{S}$ is the feasible space of states, namely $S_t \in \mathcal{S}, \forall t$.

### 4.1.2. Action Space

In fact, the action of the UAV is to decide the next hovering access point to serve mobile devices based on the current environment state and its strategy. To simplify the analysis and reduce the complexity, we consider the finite and fixed access points, and the UAV needs to choose one of them to collect tasks of devices in the coverage to achieve the maximum long-term reward. Thus, the action space can be expressed as

$$A_t = \{a_t \mid a_t = (x_0(t), y_0(t))\}. \tag{19}$$

In time slot $t$, the UAV moves to the fixed access point $(x_0(t), y_0(t))$. After that, the UAV collects the information of the devices within the service area of the access point, then forwards them to the BS and updates the AoIs of devices at the same time.

### 4.1.3. Reward Function

In this paper, we focus on simultaneously minimizing the AoI of devices and energy consumption of the UAV, such that the system rewards takes both factors into account. In time slot $t$, the system reward is given by

$$r_{t+1} = -\omega_1 Z(t) - \omega_2 e_{total}(t). \tag{20}$$

Accordingly, the long-term expected reward $G_t$ is the sum of discounted rewards in the following time slots and is written as

$$G_t = \sum_{k=0}^{\infty} \gamma^k r_{t+k+1}, \tag{21}$$

where $\gamma \in [0, 1]$ is the discount rate that evaluates the importance of present and future returns.

### 4.2. Double Deep Q Learning Network

To maximize the long-term expected reward $G_t$, it is necessary to find an optimal strategy $\pi^*$, which indicates the probabilities mapping from any state of $\mathcal{S}$ to an action of $A$. In general, the state-action value function, namely $Q$ function, $Q(s, a)$, is exploited to evaluate the quality of taking action $a$ with a given state $s$. Given the strategy $\pi$, the $Q$ function is defined as

$$Q_\pi(s, a) = E_\pi[G_t \mid S_t = s, A_t = a], \tag{22}$$

where $G_t$ is defined in (21). Once the $Q$ function $Q_\pi(s, a)$ is derived, it is easy to determine the optimal strategy. While it is usually difficult for the $Q$ function due to the complex environment with massive states and actions. By exploiting reinforcement learning, the agent can approximate $Q(s, a)$ through iterative interactions with the surrounding environment. Through experiments and feedback, the agent obtains some data samples in the form of $(s_t, a_t, s_{t+1}, r_t)$, which is continuously derived based on the iterative interactions. Given these updated samples, the agent conducts the training procedure to approximate the $Q$ function and then adapts its strategy. With deep Q learning, the $Q$ function is presented by $\theta$ parameterized DQN, i.e., $Q_\pi(s, a; \theta)$, which can greatly relieve the complexity due to massive states and actions.

In this paper, we apply a deep Q learning algorithm with an empirical replay to explore the unknown environment. To achieve the maximum long-term reward, the agent adjusts its strategies based on the feedback of the system reward by trying different actions and then strengthening the corresponding actions until the best results are achieved.

In deep Q learning, the DQN is used to approximate the $Q$ function $Q(s, a; \theta)$, where $\theta$ represents the network weight. In [51], DQN uses two-depth neural networks to approximate the Q value. One is the prediction network, where the input is the current state action pair $(s_t, a_t)$ and the output is the predicted value, that is $Q_p^{DQN}(s_t, a_t; \theta)$. Another one is the target network, where the input is the next state and the output is the maximum Q value of the next state-action pair. The target value of DQN is given by

$$Q_t^{DQN}(s_t, a_t; \theta^-) = r_t + \gamma \max_a Q(s_{t+1}, a, \theta^-). \tag{23}$$

Although DQN can quickly make the Q value approach to the possible optimal value, it may occur that the update of the parameters is too late and cause overestimation, which means the obtained training model suffers a big deviation, and leads to the final strategy is not optimal, but sub-optimal [51]. To solve it, the DDQN algorithm eliminates the overestimation issue by decoupling the selection of the target Q value action and the calculation of the target Q value. The difference with DQN is that the target value in DDQN is defined as

$$Q_t^{DDQN}(s_t, a_t; \theta^-) = r_t + \gamma Q(s_{t+1}, \arg\max_a Q(s_{t+1}, a; \theta); \theta^-). \tag{24}$$

*4.3. Training Process*

In this subsection, we present the detail of the training procedure. In the DRL framework, we take the UAV as an agent to explore the unknown external environment. As shown in Figure 3, the $Q$ function $Q(s, a|\theta)$ is characterized by DNN parameter $\theta$. The agent interacts with the environment and learns path planning strategy constantly. At time slot $t$, the agent takes action $a_t$ at the current environment state $s_t$. After that, the agent receives a feedback reward $r_t$ and the environment evolves to the next state $s_{t+1}$ with some probability. Therefore, the training samples are generated in the time slot $t$ as $o(t) = (s_t, a_t, s_{t+1}, r_t)$. This procedure continues until the environment sends the termination state. The generated training samples are stored in the relay memory $O(t) = \{o(t - O + 1), \ldots, o(t)\}$, where $O$ is the buffer length. In each episode, a small batch of empirical K is randomly extracted from memory and updated with a variant $\theta$ based on the stochastic gradient descent (SGD) method, which is named as an empirical replay. According to this manner, the parameters in the evaluation network are trained by minimizing the following loss function

$$L(\theta) = E\left[(\Omega_t - Q(s_t, a_t; \theta))^2\right], \tag{25}$$

where the $\Omega_t$ is the target value, and the target network can be estimated as follows:

$$\Omega_t = \begin{cases} r_t, & \text{if } s_{t+1} \text{ is terminal} \\ r_t + \gamma Q(s_{t+1}, \arg\max_a Q(s_{t+1}, a; \theta); \theta^-), \text{otherwise} \end{cases} \tag{26}$$

In above expression, $s_{t+1}$ is the terminal state in which the residual energy of the UAV is lower than the threshold.

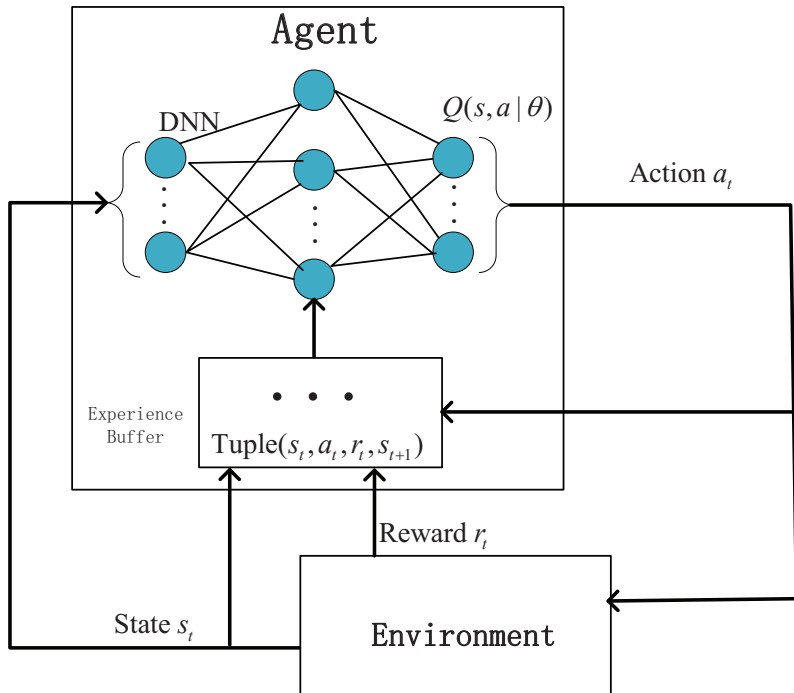

**Figure 3.** Training process.

Then, we propose a DDQN-based method to find the strategy in which $\varepsilon$-greedy exploration strategy is used to balance the proportion of exploration and development in the training process. With $\varepsilon$-greedy exploration strategy, the agent randomly selects actions with the probability of $\varepsilon$. Otherwise, the action with the maximum estimated value is selected with probability $1 - \varepsilon$. A small batch of $K$ experience is extracted from the playback memory to update the weight of the evaluation network $\theta$. By substituting $K$ experience, we can calculate the loss function as $L(\theta) = \frac{1}{K} \sum_{k=1}^{K} (\Omega_k - Q(s_k, a_k; \theta))^2$. The weight is updated by $\theta = \theta + \nabla_\theta L(\theta)$. The gradient can be calculated as $\nabla_\theta L(\theta) = \frac{2}{K} \sum_{k=1}^{K} (\Omega_k - Q(s_k, a_k; \theta) \nabla_\theta Q(s_k, a_k; \theta))$. Note that each episode ends when the UAV battery is exhausted, and then the new episode begins until the total number of episodes equals the threshold of episodes $K_{\max}$.

The goal of DRL is to continuously adjust the network parameters and learn the path planning policy $\pi^*$ (a mapping from the state in its space $S$ to the probability of selecting each action among the action space $A$) according to the continuous interaction between the agent and the environment in Figure 3, to maximize the long-term expected reward $G_t$ (i.e., Equation (21)). Algorithm 1 summarizes the training process to find the convergent policy $\pi^*$. The initialize procedure is in Line 1 to set the initial energy, the Q-network parameter. Line 2 indicates the episode of the whole training process. Line 3, gives the initialize procedure of the episode. Then, in Line 5, the agent (UAV) selects an action $a_t$ based on the current state $s_t$, and Line 6 and 7 are associated with two different selection strategies. In Line 8, the action is executed, the agent obtains a reward $r_t$, and the environment is transited to another state $s_{t+1}$. The current interaction sample is then stored in the memory Q in Line 9. From line 10 to line 14, the agent extract K mini-batch samples from the memory, and update/refine its policy based on a deep neural network. After multiple episode training processing, the above algorithm can finally derive a converged policy $\pi^*$. The training procedure is offline, and then the strategy $\pi^*$ is to instruct the UAV to approach the maximum long-term reward during online testing.

---

**Algorithm 1** DDQN-based freshness-aware path planning

---

1: **Initialize:** The relay memory $O(0) = 0$, the initial energy $E(0) = E_0$, deep Q-network weight $\theta^- = \theta$, $\gamma$, and $\varepsilon$.
2: **for** episode $j = 1, 2, \ldots \ldots, K_{\max}$ **do**
3:    Set the time slot index $t = 0$, and generate the initial state $s_0$; Generate the initial position of the UAV and devices.
4:    **while** $E(t) > 0$ **do**
5:       Take an action $a_t$ with a given state $s_t$, based on $\varepsilon$-greedy policy:
6:       Case i: Randomly select an action with the probability $\varepsilon$.
7:       Case ii: Select action: $\arg\max Q(s_{t+1}, a)$.
8:       Obtain a reward $r_t$, and then the state is transited to $s_{t+1}$;
9:       Store the tuple sample $(s_t, a_t, r_t, s_{t+1})$ into the memory $O$;
10:      **if** $O$ is full **then**
11:        Randomly extract $K$ mini-batch samples from memory $O$;
12:        Compute the loss function $L(\theta)$
13:        After a fixed interval,update $\theta^-$ as $\theta^- = \theta$
14:      **end if**
15:      Update the energy status of the UAV as $E(t+1) = E(t) - e_{total}(t)$ and $t = t + 1$.
16:    **end while**
17: **end for**
18: **Output:** Policy $\pi^*$

---

## 5. Experiments and Results

In this section, we evaluate the achieved performance of the proposed DDQN-based freshness-aware path planning scheme for UAV-assisted edge computing networks through the Monte-Carlo simulation. Unless otherwise stated, the simulation parameters are presented in Table 2. In the DDQN framework, we exploit a four-layer fully-connected neural network that consists of one input layer, two hidden layers, and one output layer. The number of neurons in the hidden layers are 120 and 80, respectively, and the Relu function, $f(x) = max(0, x)$, is used as the activation function of all hidden layers. Meanwhile, the network parameter $\theta$ is updated by an Adam optimizer with a learning rate of 0.00001. To explore the environment, we set $\varepsilon = 0.1$, the experience playback buffer size is $|B_m| = 2.5 \times 10^5$. To compare the performance of different algorithms, the average test results are taken from 20 numerical simulations, each of which contains 100,000 steps.

**Table 2.** Simulation parameters.

| Symbol Notations | Parameter |
| --- | --- |
| Number of access points | $M = 9$ |
| Initial energy of UAV | $E_0 = 5 \times 10^4$ J |
| Flying speed of UAV | $V = 25$ m/s |
| Reference Path loss | $\rho_0 = -50$ dB |
| Hovering power consumption | $P_h = 100$ W |
| Flying power consumption | $P_f = 150$ W |
| Relay forwarding time | $t_u = 0.5$ s |
| System bandwidth | $B = 3$ MHz |
| Noise power spectral density | $N_0 = 4.0 \times 10^{-12}$ W/Hz |
| Communication time of the UAV | $T_s = 1$ s |
| Maximum angle from UAV and the edge of its coverage area | $\alpha = \pi/6$ |
| Probability of task generation | $p = 0.6$ |

In Figure 4, the achieved AoI of the proposed DRL-based freshness-aware path planning scheme is compared with that of the random scheme, where three kinds of deep reinforcement algorithms are investigated, including DQN, DDQN, Dueling DQN. As a variant of DQN, Dueling DQN decomposes the $Q$ function $Q(s, a)$ into the value function $V(s)$ and the advantage function $A(s, a)$ [52]. The random path planning scheme selects

the access point in a random manner. By varying the number of devices $N$ from 10 to 60, we observe that the sum AoI of all devices under the DRL-based path planning schemes is much better than that of the random path planning scheme. With the benefit of structure improvement of DQN, DDQN, and Dueling DQN performs better, where the DDQN is best. It is shown that DDQN can compensate for the overestimation of Q value of DQN. Therefore, Figure 4 proves the superiority of the DDQN-based path planning of the $\varepsilon$-greedy strategy in this scenario.

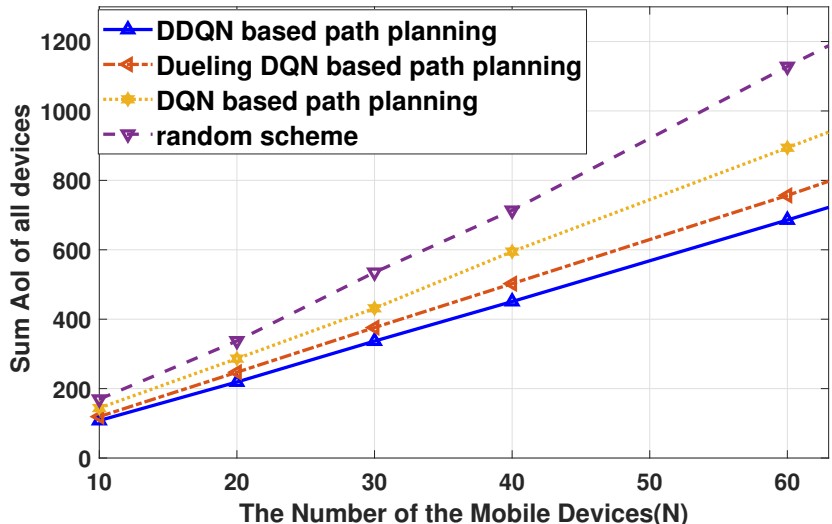

**Figure 4.** Achieved AoI of proposed path planning scheme with a different number of mobile devices, and $v = 2$ m/s, $H = 15$ m.

Figure 5 evaluates the effect of learning rate on the convergence of reward, where $N = 30$ and the learning rate $\beta$ are set to 0.01, 0.00001, 0.000001, respectively. As is shown, the convergence of the DDQN-based path planning can be guaranteed. When $\beta = 0.01$, there exists a reward loss, which is because the adjustment range of the neural network parameter is too large. The system reward in the cases of $\beta = 0.00001$ and $\beta = 0.000001$ are almost the same. While, the scheme converges much faster in the case of $\beta = 0.00001$, such that we set the learning rate $\beta = 0.00001$ in the following parts. After the neural network outputs different Q values, what kind of strategy to choose the action of the current state will directly affect the reward. As shown in Figure 6, we compare three selection strategies, including the $\varepsilon$-greedy strategy of $\varepsilon = 0.1$ and $\varepsilon = 0.3$, the GreeyQ strategy, and the BoltzmannQ strategy. GreeyQ strategy is a greedy strategy to choose the maximum Q value all the time. BoltzmannQ strategy establishes a probability law according to Q value and returns a random selection behavior according to this law. We can observe that the average reward of $\varepsilon$-greedy strategy of $\varepsilon = 0.1$ is better than $\varepsilon$-greedy strategy of $\varepsilon = 0.3$ and GreeyQ strategy, while the average reward of BoltzmannQ strategy is the smallest. As the GreeyQ policy is a deterministic strategy, that is, the probability of the UAV is only 1 at the next fixed access point that maximizes the action value function, and the probability of selecting other fixed access points is 0. But this will make the UAV only rely on past experience to determine its own service path and lose the ability to continue to explore and develop. Therefore, for the agent of this system, it is more reasonable to maintain a certain probability of utilization and exploration.

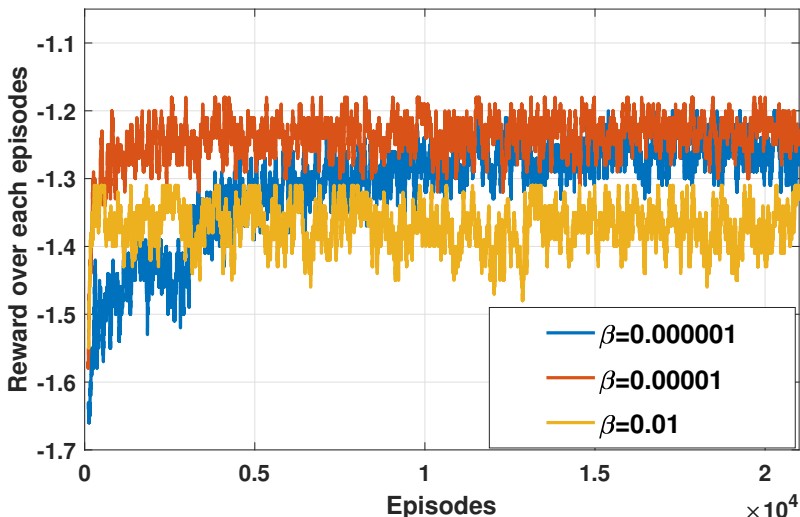

**Figure 5.** Effect of learning rate on the convergence performance of reward.

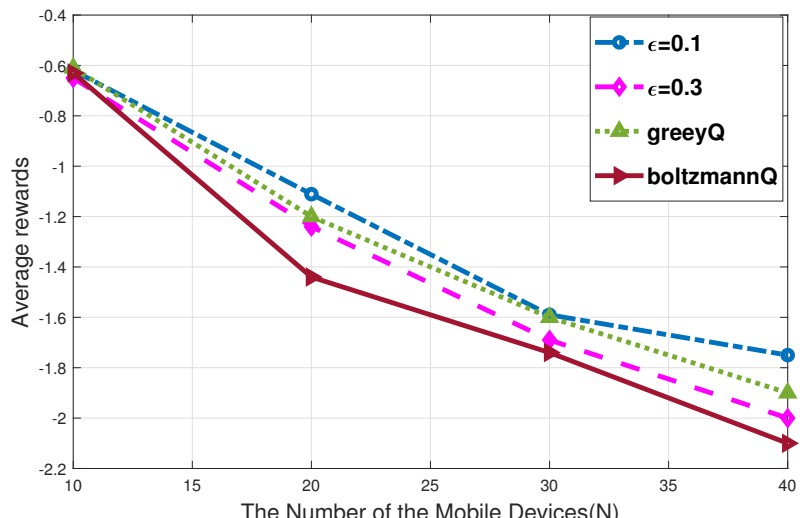

**Figure 6.** In the case of $v = 2$ m/s, $H = 15$ m, the impact of different selection strategies on system reward.

We can see from Figure 7 that for idle mobile device queues, different task arrival probabilities have a certain impact on the update of the device's AoI. We consider the impact of different task generation probability $p$ on the average AoI when $N = 10, 20$, 30, 40, and 60, respectively. When the probability of task arrival is 0.1, the average AoI is maximum. When the probability of task arrival is 0.3, 0.6, the average AoI decline accordingly. However, when the probability of task arrival is 0.9, it is not much different from the probability of task arrival of 0.6. At a relatively small probability $p$, fewer devices are required for the task. At this time, the UAV does not need to collect more data packets to the BS for unloading processing, which should result in a greatly reduced device freshness. When the generation probability of device queue tasks is relatively high, the UAV needs to arrive at the target point in time to process the data packet. After completing the task unloading, the AoI of the devices can be reduced in time. However, the data processing capacity and scope of UAVs is limited; even if many devices on the ground have service requirements, the system can not be completely timely processing. Therefore, when the larger task arrival probability $p$, the AoI of devices may not reduce significantly.

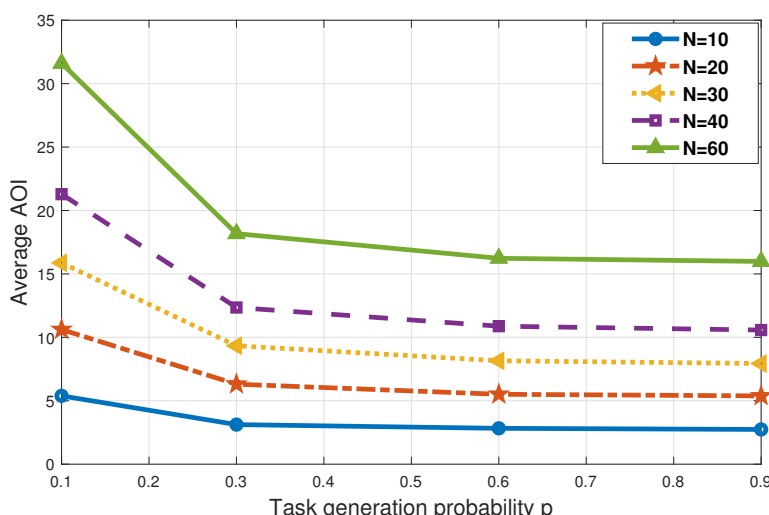

**Figure 7.** In the case of $v = 2$ m/s $H = 15$ m, the impact of different task generation probability $p$ on average AoI.

As can be observed from Figure 8, the higher the speed, the bigger the average AoI. The average AoI is the smallest when the position of the mobile device is fixed ($v = 0$ m/s). This is due to the slower the location of the mobile device changes, and the $\varepsilon$-greedy exploration strategy of the DRL algorithm makes it easier to learn the optimization strategy, which makes it easier for the UAV to quickly collect the task data of the device and forward it to the nearby BS in time. Therefore, when the speed of the mobile device slows down, the average AoI of the mobile device is reduced. Next, we assume that the battery power of the UAV is sufficient for a period of service work and consider the same number of steps in each episode. Figure 9 shows that the higher the speed of the UAV, the smaller the corresponding average AoI. This is because, in the process of working, UAVs sacrifice greater energy consumption to reduce flight time, timely processing of devices' data, and maintain information freshness.

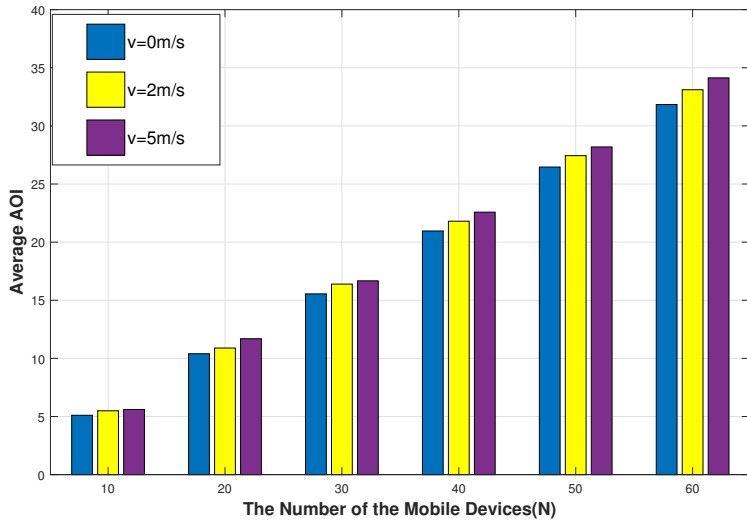

**Figure 8.** The impact of different speeds of mobile devices on average AoI.

The service path trajectory of the UAV is presented in Figure 10 for a 60 m × 60 m communication area, with $v = 2$ m/s, $H = 15$ m and $M = 4$. The numbers 1 to 8 in the figure are the serial numbers of the devices. The time slot is $t = 3, 5, 8, 11$, and it can be observed that the UAV with limited energy can process task data from N mobile devices in

the scene through the route of C→B→D→B. We can observe that in each time slot, UAVs do not necessarily stay at the original access point or fly at the nearest access point but are likely to fly to distant access points to process other mobile device data. In the environment state of $t = 3$, the UAV is in area C. Although there are more mobile devices with service requirements in area D, the UAV preferentially selects the nearby area B as the mobile devices with device numbers 2 and 3 for data processing in the next time slot. Under the framework of the reinforcement learning algorithm, UAV pays attention to long-term system benefits rather than short-term limited feedback. Finally, the evolution of AoI of User1 and User2 in an episode is presented. As shown in Figure 11, we can observe that the AoI of devices increases with fly time and communication time slot. Only when the UAV accesses the corresponding fixed access point and assists the BS in unloading does the AoI decreases accordingly. Therefore, the path planning of the UAV affects the update of the specific devices' AoI.

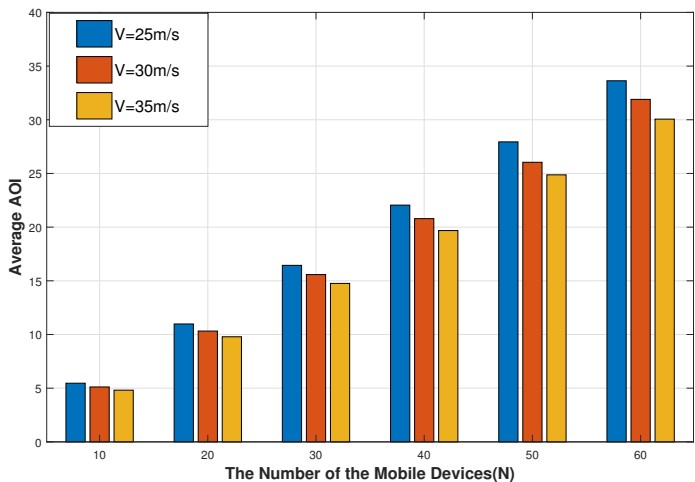

**Figure 9.** The impact of different speeds of the UAV on average AoI.

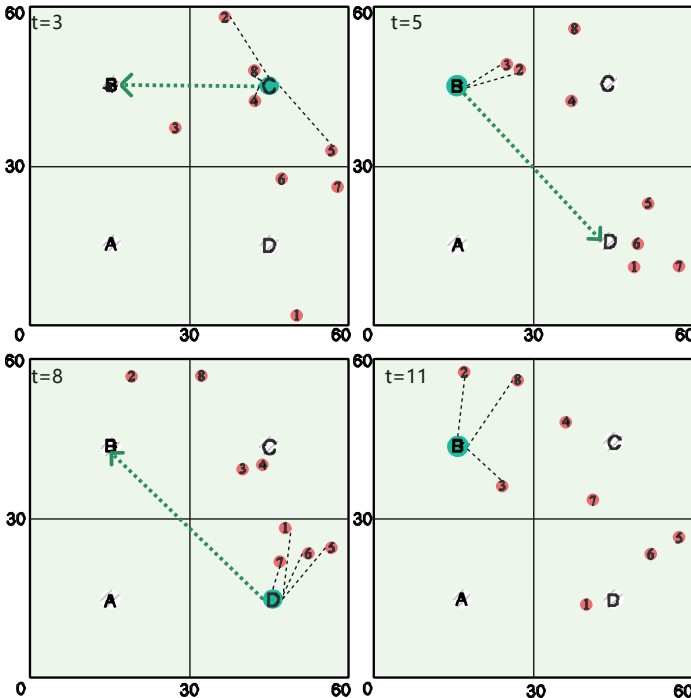

**Figure 10.** Service path of the UAV with four steps.

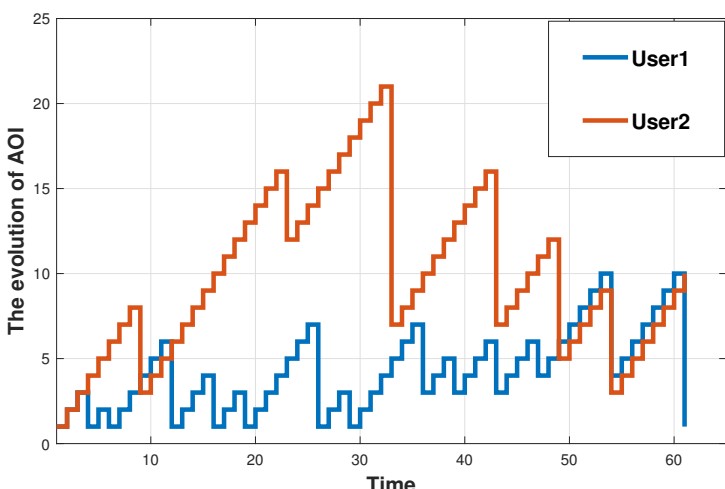

**Figure 11.** AoI evolution of two mobile devices.

## 6. Discussion

The intelligent and freshness-aware path planning for an UAV-aasisted MEC network has been studied in this paper. In the future, the resource management and path planning for the UAV-assisted MEC networks under practical propagation scenarios can be further studied, such as combing LoS and NLoS links and imperfect or no channel state information (CSI). Meanwhile, some emerging techniques can be incorporated. For example, the intelligent reflecting surface (IRS) technique can reconfigure the wireless propagation environment to achieve a flexible and efficient communication network. Non-orthogonal multiple access technique shows the benefits of improving the spectral efficiency and network connectivity. Physical layer security technique can be exploited to realize an energy-efficient and secure communication network. Furthermore, more complicated network scenarios need to be investigated, e.g., multi-UAV or multi-cell communication networks or large-scale UAV-assisted communication networks with massive IoT devices.

## 7. Conclusions

In this paper, we have studied a UAV-assisted MEC network with device mobility, where UAVs with limited energy can assist mobile devices in offloading their tasks to the BS. UAV brought a wide range of flexible services, leading to path planning problems. Our goal was to minimize AoIs of mobile devices and the energy consumption of UAV work. We applied the DDQN algorithm to plan the service path of the UAV intelligently. Simulation results have validated the effectiveness of the proposed freshness-aware path planning scheme and the effects of the moving speed of devices and the UAV on the achieved AoI. Finally, we have given the process of AoI evolution and the service trajectory of the UAV.

**Author Contributions:** Conceptualization, Y.P. and Y.L.; methodology, Y.P. and Y.L.; writing—original draft preparation, Y.P.; writing—review and editing, Y.L., H.Z. and D.L.; supervision, Y.L. and H.Z. All authors have read and agreed to the published version of the manuscript.

**Funding:** This work was supported by the National Natural Science Foundation of China under grant No. 61901180, Natural Science Foundation of Guangdong Province under Grant No. 2019A1515011940, No. 2022A1515010111, Science and Technology Project of Guangzhou under grant No. 202002030353, Basic and Applied Basic Research Project of Guangzhou under grant No. 202201010574, and Science and Technology Development Fund of Macau under grant No. 0018/2019/AMJ, No. 0110/2020/A3 and No. 0029/2021/AGJ.

**Data Availability Statement:** Not applicable.

**Conflicts of Interest:** The authors declare no conflict of interest.

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
