# Peer review of "Deep Reinforcement Learning Based Freshness-Aware Path Planning for UAV-Assisted Edge Computing Networks with Device Mobility"

_remotesensing, doi:10.3390/rs14164016_

Round 1
Reviewer 1 Report
The paper is describing the use of a deep learning network trained using a reinforcement learning methodology for optimizing a mobile network using UAV nodes to improve the communications for mobile and fixed nodes. The idea is interesting and the paper demonstrates that the approach is capable of providing good performance compared to some other schemes taken from the literature.
Here are my questions/comments about this current work:
1. There are several places where the language of the paper could be improved. I would recommend another proofread to correct some English grammar issues.
2. The description of Algorithm I on pages 13 could be clarified. The current algorithm states that it outputs the Optimal Policy Pi^{*} but this policy is not referenced directly anywhere in the algorithm description above. It would be nicer to have some explicit description of how the policy Pi^{*} is generated here. I am aware that this is in the main text but it should be reference here.
3. The reinforcement learning is an important part of this work since the reward is often not known at a given time only if the system is performing well or not at a given time. The paper claims that the 'optimal' policy is returned. What is the criteria of optimality used? Is the policy optimal with respect to the estimated reward function or the unknown reward function?
4. With online learning from data sequences generated from system (simulated or real) observations taken over time, it is important to give some justification that the parameters of the system are proper. That is, are the system parameters for the simulated network in this paper in a steady-state distribution when the learning algorithm is applied? Some parameters of interest would be the locations of the mobile nodes, the velocity of the mobile nodes, and the traffic parameters of the generated communications. You would also need to have some justification for these parameter sequences being ergodic; that is, can their statistical parameters be inferred from observations of a data sequences? Answering these questions in the affirmative would allow the authors to justify that their system can learn the 'optimal' path planning algorithms and that the reported results were not just a fluke of individual random selection of simulation parameters. My belief is that the answer to all of these questions is in the affirmative, but the authors need to include some mathematical justification. Some statements of how the simulations were demonstrated to be in steady-state before performance was measured would go a long way in satisfying this.
I think that there is good work here that many people will find useful for their studies. With these questions above answered, I think that this work is worthy of publication.
Reviewer 2 Report
This paper presents an investigation on UAV assisted mobile edge computing (MEC) in which the UAV serves as a relay between the edge node and the base station. The authors have aimed to minimize the Age of Information (AoI) of mobile devices. DDQN and DRL have been used to analyze the moving speed of the devices.
The paper is well-written and organized. Several results have been presented to support the argument however, I am not convinced with the model itself. To me, bringing in the UAV between the BS and the edge node is a flawed theory. This will simply be like fog computing scenario and instead of a fog node, a UAV is replaced. Moreover, bringing in the UAV will bring in several issues of reliability and availability. Last but not least, this will add delay and the proposed model will not be suitable for time critical applications.
I believe, this has been the reason the authors have been unable to relate the proposed model with a real world scenario.
I will suggest the authors to revisit their theory and support it by sound references and relate it with real world problems.
Reviewer 3 Report
This paper studies a UAV-assisted mobile edge computing 2 (MEC) network, where the UAV is deployed as a relay station to collect computing tasks from devices and then forward them to a faraway base station (BS). Meanwhile, to satisfy the freshness requirement of IoT applications, the recently proposed performance metric, namely age of information (AoI), is incorporated in this paper. To this end, the authors formulate an optimization problem to simultaneously minimize the AoIs of mobile devices and the UAV's energy consumption by planning the UAV's service path, where the dynamic and random moving of all devices are taken into account. Concerning the dimension explosion issue, they explore a double deep Q-learning network (DDQN) algorithm based on deep reinforcement learning (DRL) framework to realize intelligent service path planning. Simulation results validate the effectiveness of the proposed freshness-aware path planning scheme and the effects of the moving speed of devices and the UAV on the achieved AoI.
Overall this paper is written and organized well. However, I have the following comments that must be addressed before considering this paper.
1) The abstract should be written in a more technical way. For instance, the abstract should begin with a brief but precise statement of the problem or issue, followed by a description of the research method and design, major findings, and conclusions.
2) Some related work is missing in the related work subsection. For example, Energy efficient UAV flight path model for cluster head selection in next-generation wireless sensor networks, sensors 2021. All the related work must be carefully searched, studied, and reported. Then, the main contribution of this can be revised properly.
3) In the contributions, the authors claim that we consider the mobility of the IoT devices and UAVs, and how to get channel state information (CSI) in such scenarios? as its conditions change quickly. Do you consider perfect CSI? which is very challenging. In such a scenario, it is better to consider imperfect CSI. I recommend authors study some research works that consider imperfect CSI and see how the system performance can be affected if CSI has errors. Here I recommend some recent studies, such as Energy efficiency optimization for backscatter enhanced NOMA cooperative VEX communications under imperfect CSI; Energy-efficient backscatter aided uplink NOMA roadside sensor communications under channel estimation errors. The most important things are that these studies consider imperfect CSI and energy efficiency. Therefore, the authors can study and report the effect of imperfect CSI on the system's energy efficiency in the revised manuscript.
4) This work uses the reinforcement learning technique; why do they not consider other AI techniques? What is the advantage of this technique for the current model? It is suggested to add motivation for using this AI technique. The following paper might be useful to study and report: Reinforcement learning in blockchain-enabled IIoT networks: a survey of recent advances and open challenges.
5) To see the performance of the proposed technique, it is better to directly compare it with any recent work in the literature.
6) In conclusion, the authors can add some potential research issues and directions for the reader of this work.
7) How to improve the security issues in such a network. Recently researchers have claimed that intelligent reflecting surfaces are a promising technology for energy efficient and secure transmission in future wireless networks. How IRS could be integrated into this model can affect energy consumption and security problems. Recent work, i.e., Opportunities for physical layer security in UAV communication enhanced with intelligent reflective surfaces, has addressed all these issues.
8) Some typos and grammar errors can be easily detected in this work. Proper proofreading is required. Also, each and every equation in this paper need to be properly defined in the first place.
Round 2
Reviewer 2 Report
My comments have been addressed. The paper can be accepted.
Just improve the quality of the figures.
Author Response
The authors appreciate the reviewer’s recognition of the contributions of our work in this paper, and we would like to thank the reviewer for the constructive comments, which help us to improve the quality of the paper.
We have rechecked all the figures in the manuscript, reshaped most of them (including Figure 1, 2,4,5,6,7,10,11), and incorporate more details to clearly present the system model, the transmission scheme, and the simulation results. For the details, please refer to Page 5, 6, 13,14,17,18 of the revised manuscript.

Reviewer 3 Report
Thank you so much for addressing all of my comments. This article looks very good now. I have only one comment in this round. Please explain how edge computing can be efficient in data/task offloading. Some latest and good review papers on this topic can be helpful to study and report.
1) Optimal resource allocation and task segmentation in iot enabled mobile edge cloud, A Mahmood, Y Hong, MK Ehsan, S Mumtaz IEEE Transactions on Vehicular Technology 70 (12), 13294-13303, 2021. 2) RL/DRL meets vehicular task offloading using edge and vehicular cloudlet: A survey, J Liu, M Ahmed, MA Mirza, WU Khan, D Xu, J Li, A Aziz, Z Han IEEE Internet of Things Journal 9 (11), 8315-8338, 2022.The paper can be accepted after incorporating these changes.
Author Response
Thank you so much for your constructive comments, which help us to improve the quality of the paper.
In cloud computing framework, the devices are served by the remote cloud centers, which usually causes long latency due to the long propagation distance during the data exchange. Instead, mobile edge computing (MEC) integrates the concept of cloud computing into the network edge of mobile networks, i.e., radio access network, which is aligned with a key characteristic of next-generation networks that the information is increasingly generated locally and consumed locally. Therefore, by offloading the computation tasks from mobile devices to the MEC servers, the quality of computation experience, including energy consumption and execution latency, can be greatly improved.
In the revised manuscript, we add a sentence in the first paragraph of the introduction section to explain how edge computing can be efficient in data/task offloading:
The MEC offers the computing capability at the network edge in close proximity to end devices, i.e., radio access network, such that the large distance transmission is no longer necessary, and the information is generated locally and consumed locally. Thus, the quality of service is improved due to a small energy consumption and execution latency [3,4].
- Mahmood A.; Hong Y.; Ehsan M. K.; Mumtaz S. Optimal Resource Allocation and Task Segmentation in IoT Enabled Mobile Edge Cloud. IEEE Transactions on Vehicular Technology. 2021, 70(12), 13294-13303.
- Liu, J.; Ahmed, M.; Mirza, M. A.; Khan, W. U.; Xu, D.; Li, J.; Han, Z. RL/DRL meets vehicular task offloading using edge and vehicular cloudlet: A survey. IEEE Internet of Things Journal. 2022, 9(11), 8315-8338.
